# A Novel Reperfusion Strategy for Primary Percutaneous Coronary Intervention in Patients with Acute ST-Segment Elevation Myocardial Infarction: A Prospective Case Series

**DOI:** 10.3390/jcm12020433

**Published:** 2023-01-05

**Authors:** Ji-Fang He, Yi-Xing Yang, Jiang-Yuan Li, Lu Liang, Li Xu, Yu Liu, Zong-Sheng Guo, Qi Yang, Tao Jiang, Xiang-Min Lin, Xin-Chun Yang, Mu-Lei Chen, Pi-Xiong Su, Jiu-Chang Zhong, Le-Feng Wang

**Affiliations:** 1Heart Center and Beijing Key Laboratory of Hypertension, Beijing Chaoyang Hospital, Capital Medical University, No. 8, Gongti South Road, Chaoyang District, Beijing 100020, China; 2Department of Radiology, Beijing Chaoyang Hospital, Capital Medical University, No. 8, Gongti South Road, Chaoyang District, Beijing 100020, China

**Keywords:** percutaneous coronary intervention, ST-segment elevation myocardial infarction, ischemia–reperfusion injury, microvascular obstruction

## Abstract

Background: Ischemia reperfusion injury (IRI) remains a major problem in patients with acute ST-segment elevation myocardial infarction (STEMI) undergoing primary percutaneous coronary intervention (PCI). We have developed a novel reperfusion strategy for PCI and named it “volume-controlled reperfusion (VCR)”. The aim of the current study was to assess the safety and feasibility of VCR in patients with STEMI. Methods: Consecutive patients admitted to Beijing Chaoyang Hospital with STEMI were prospectively enrolled. The feasibility endpoint was procedural success. The safety endpoints included death from all causes, major vascular complications, and major adverse cardiac event (MACE), i.e., a composite of cardiac death, myocardial reinfarction, target vessel revascularization (TVR), and heart failure. Results: A total of 30 patients were finally included. Procedural success was achieved in 28 (93.3%) patients. No patients died during the study and no major vascular complications or MACE occurred during hospitalization. With the exception of one patient (3.3%) who underwent TVR three months after discharge, no patient encountered death (0.0%), major vascular complications (0.0%), or and other MACEs (0.0%) during the median follow-up of 16 months. Conclusion: The findings of the pilot study suggest that VCR has favorable feasibility and safety in patients with STEMI. Further larger randomized trials are required to evaluate the effectiveness of VCR in STEMI patients.

## 1. Introduction

Primary percutaneous coronary intervention (PCI) remains the default therapy for patients with ST-segment elevation myocardial infarction (STEMI), owing to its capacity for timely reopening of the infarcted artery [1]. However, abrupt reperfusion of coronary blood flow can induce myocardial and vascular injury. This phenomenon, known as ischaemia–reperfusion injury (IRI), is reported to account for up to 50% of final myocardial infarct size, and may counteract the beneficial effects of primary PCI [2].

Postconditioning using a sequence of ischaemia–reperfusion episodes induced by repeated cycles of balloon-catheter inflation and deflation immediately after reopening of the occluded vessel has been proven a valid alternative to protect the myocardium against reperfusion injury, according to animal models and preclinical studies [3,4,5]. Unfortunately, the results of studies evaluating the effects of postconditioning on surrogate markers (including infarct size, edema, and microvascular obstruction) in humans have been inconsistent, with several studies showing positive effects [6,7] and others indicating neutral or negative effects [8,9,10]. Moreover, recent randomized controlled trials such as POST and DANAMI-3 found that routine postconditioning during primary PCI failed to improve clinical outcomes in patients with STEMI at both short- and long-term follow-up [11,12]. These disappointing results might be partially attributed to the major shortcomings inherent in postconditioning [13,14,15]. Although coronary blood flow is intermittently transfused, it cannot prevent the distal vascular bed and myocardium from incurring injury by abrupt blood reperfusion when the balloon deflates. Furthermore, the patient’s arterial blood may contain a higher volume of inflammatory factors at the STEMI setting, and repeated blood flow lash might cause additional damage to the vascular endothelium and myocardium. Moreover, when postconditioning protocol is performed after pre-dilatation, IRI might have already occurred after wire passing or after pre-dilatation. Despite being easy to perform, repeated inflation and deflation of the balloon can induce damage to the coronary endothelium and increase the risk of thrombus dislodgment. Modified postconditioning or a brand-new methodology is required to overcome these difficulties.

Interestingly, earlier studies using surgical or nonsurgical models of ischemia reported that gradual restoration of coronary blood flow by slowly reopening the occluded infarct-related artery to achieve “gentle reperfusion” could significantly reduce reperfusion injury [16,17]. However, early reports of these protective effects afforded by gradual reperfusion did not capture the attention of physicians and to date no study has yet investigated the effectiveness of gradual reperfusion in clinical settings, to the best of our knowledge. In addition, despite the advantages of gradual reperfusion, it cannot overcome the limitations resulting when vascular endothelial cells and cardiomyocytes are subjected to an onslaught of coronary blood flow with high concentrations of inflammatory factors at the beginning of reperfusion restoration, and it is difficult to control effectively the volume and speed of coronary flow during the reperfusion process.

Based on the concept of postconditioning and gradual reperfusion, and using the dual catheter technique [18], we modified the conditions of reperfusion and proposed a novel strategy referred to as “volume-controlled reperfusion (VCR)”. Basically, this strategy enables accurate intermittent slow distal reperfusion using a manual intra-aspiration catheter after suspending the antegrade flow using a prophylactic inflated balloon. Theoretically, VCR could overcome the limitations of postconditioning and gentle reperfusion. First, intermittent reperfusion via a manual aspiration catheter at controlled volume and speed can avoid abrupt refusion and high intracoronary pressure. Second, heparin as an auxiliary but necessary agent for perfusate might reduce the risk of IRI by inhibiting the abnormal aggregation and adhesion of leukocytes and platelets, preventing the formation of microthrombus, while reducing local inflammation and oxidative stress response [19,20]. Third, the technique can avoid repeated inflation and deflation of the balloon during post-conditioning, which can reduce the damage to the coronary endothelium and the risk of thrombus dislodgment. The objective of this pilot study was to evaluate the safety and feasibility of VCR in patients with STEMI undergoing PCI.

## 2. Methods

### 2.1. Study Population

Consecutive patients admitted to Beijing Chaoyang Hospital with STEMI were prospectively enrolled between 1 August 2020 and 1 December 2021. The inclusion criteria were: (1) age between 18 and 80 years; (2) first STEMI attack; (3) presenting within 12 h of symptom onset; (4) proximal occlusion of the dominant coronary artery; (5) thrombolysis in myocardial infarction (TIMI) grade 0 or 1 flow in the infarct-related artery (IRA). The exclusion criteria were: (1) previous myocardial infarction; (2) previous coronary bypass surgery; (3) unconsciousness or cardiogenic shock (≥Class C); (4) mechanical complications; (5) left main lesion; (6) evidence of coronary collaterals (Rentrop grade > 1) to the region at risk; (7) severe renal or liver failure; (8) ongoing malignant process. The study protocol was approved by the Ethics Committee of our institution. All subjects provided written informed consent.

### 2.2. Procedural Protocol

Loading doses of aspirin 300 mg, clopidogrel 300 mg, and intravenous heparin at 70–80 U/Kg were administered before the procedure. The detailed steps of the VCR procedure were as follows (Figure 1):

(1)A guiding catheter (GC1) was engaged to the target coronary artery via a transradial approach, followed by delivery of a guide wire (GW1). After the GW1 passed over the index lesion to the distal segment of target coronary artery, an appropriately sized balloon was immediately positioned at the site of the index lesion and inflated with appropriate pressure (6–12 atm) to completely occlude the forward coronary flow.(2)After ensuring reocclusion of coronary flow, a second guiding catheter (GC2) was immediately engaged into the target coronary artery through a transfemoral approach. Then, an aspiration catheter filled with contrast medium to prevent bubbles from entering the coronary arteries was advanced on a second guide wire (GW2) and positioned 15–20 mm distal to the index lesion. The balloon remained inflated throughout the entire process except for the moment when the GW2 and aspiration catheter passed through the index lesion.(3)Before reperfusion, a gentle puff of contrast via the aspiration catheter was applied to confirm the distal lumen patency. At this point, the VCR model was successfully established. Then, arterial blood was extracted via GC1 and diluted with heparin saline (3000 units of heparin to 500 mL of normal saline) at a 1:1 ratio (10 mL:10 mL) and subsequently manually infused through the aspiration catheter into the distal part of the target vessel. The perfusate infusion started at a flow rate of 20 mL/min and was subsequently adjusted based on heart rate and blood pressure measurements to maintain stable heart rate and blood pressure. If the heart rate or blood pressure dropped, the process was paused or the reperfusion speed reduced until all parameters returned to stability. Ten rounds of reperfusion were generally applied, with each round comprising 60 s reperfusion time and 15 s pause for perfusate preparation.(4)After the first reperfusion step was completed, the balloon was deflated and withdrawn to GC1 and the aspiration catheter withdrawn to GC2. Then, angiography and conventional PCI procedures were performed in sequence. Stent selection, use of glycoprotein IIb/IIIa inhibitors, or intra-aortic balloon pump (IABP) were left to the operators’ discretion.

### 2.3. Endpoints and Definitions

The feasibility endpoint was procedural success, defined as successfully establishing the VCR model (femoral and radial artery routes successfully opened; the dual guiding catheters successfully engaged to the target coronary artery; the balloon successfully positioned at the site of the index lesion; the aspiration catheter successfully advanced to the target site), maintaining stable hemodynamics during the entire infusion, and achieving a TIMI grade 3 flow at the completion of the PCI procedure.

The safety endpoints included all-cause death, major vascular complications, or major adverse cardiac event (MACE) during hospitalization or at the mid-term follow-up. Major vascular complications were defined as access-site injury or access-related vascular injury leading to death, life-threatening or major bleeding, visceral ischaemia, or neurological impairment [21]. MACE was defined as the composite of cardiac death, myocardial reinfarction, target vessel revascularization (TVR), and heart failure. Definitions of cardiac death, myocardial reinfarction, and TVR corresponded with the Academic Research Consortium criteria [22]. Heart failure was defined as any congestive heart failure [23]. Follow-up information was obtained from hospital records and by telephone interviews.

The other endpoints included ST-segment resolution (STR) at 60 min after PCI, peak levels of cardiac biomarkers after PCI, left ventricular ejection fraction (LVEF) measured by echocardiography within 3 days, and infarct size and microvascular obstruction (MVO) assessed by late gadolinium-enhanced cardiovascular magnetic resonance (LGE-CMR) within 7 days. STR was calculated as the difference in ST-segment elevation between the baseline ECG and ECG 60 min post-PCI divided by the ST-segment elevation in the baseline ECG. According to the level of STR, the patients were categorized as no STR (<30%), incomplete STR (30%–70%), or complete STR (≥70%) [24]. Infarct size was measured by manual delineation of the hyperintense myocardium on delayed enhanced images. MVO was manually calculated as the hypoenhanced region within the delayed hyperenhanced infarct region [25].

### 2.4. Statistical Analysis

Data were expressed as mean ± standard deviation (SD) or median and interquartile range (IQR) for continuous variables and as frequencies for categorical variables. All statistical analyses were performed using SPSS software version 23.

## 3. Results

Between 1 August 2020 and 1 December 2021, a total of a total of 489 patients with STEMI were admitted at Beijing Chaoyang Hospital. Of these, 30 patients were finally enrolled and received VCR treatment (Figure 2). Baseline characteristics of patients are shown in Table 1. The mean age of the overall population was 59.8 ± 12.4 years, and 26 (86.7%) patients were male. Most patients had risk factors for cardiovascular diseases, such as smoking (80%), hypertension (50%), diabetes (30%) and hyperlipidemia (20%).

Procedural and angiographic characteristics of patients are shown in Table 2. Seventeen (56.7%) patients had anterior wall infarctions (left anterior descending artery [LAD]) and thirteen (43.3%) patients had inferior wall infarctions (ten right coronary artery [RCA], three left circumflex artery [LCX]). The mean duration for VCR model establishment was 12.44 ± 2.03 min. The median VCR duration was 750 s (IQR 750, 760). The mean duration of the total PCI procedure was 44.70 ± 3.23 min. The median average aortic pressure and heart rate during VCR were 100.5 mmHg (IQR 96.00, 106.00) and 72.0 bpm (IQR 61.50, 80.25), respectively. An additional drug-eluting stent or drug-eluting balloon was administrated in 28 (93.3%) patients. Thrombosis aspiration was utilized in three (10%) patients, and glycoprotein IIb/IIIa therapy in three (10%). A total of four (13.3%) patients received intra-aortic balloon pump (IABP) implantation. Fifteen patients (50%) had multivessel disease (non-infarct-related artery with stenosis > 70%), and five finished complete revascularization during the index procedure.

Table 3, Table 4 and Table 5 shows the results for the study endpoints. The VCR model was successfully established in all 30 (100.0%) patients. All (100.0%) patients maintained stable hemodynamics during the entire infusion. Two (6.6%) patients had a TIMI grade 2 flow at completion of the PCI procedure. Therefore, procedural success was achieved in a total of 28 (93.3%) patients. No patients died and no major vascular complications or MACE occurred during hospitalization. Alongside one patient (3.3%) who underwent TVR three months after discharge, no patient encountered death (0.0%), major vascular complications (0.0%), or other MACEs (0.0%) during the median follow-up of 16 months.

With regard to the other endpoints, 21 (70%) patients exhibited complete ST-segment resolution at 60 min after PCI. The median CKMB, TNI, and BNP peaks post-PCI were 176.85 ng/mL (IQR 97.05, 289.7), 184.78 ng/mL (IQR 56.04, 258.45), and 327.00 pg/mL (IQR 126.75, 567.00), respectively. Echocardiography showed a preserved left ventricular ejection fraction (≥50%) in 27 (90%) patients. LGE-CMR was performed in a total of 16 (53.3%) patients. Median infarct size and MVO assessed by LGE-CMR were 33.77 g and 0.64 g, respectively.

## 4. Discussion

In this pilot study, we have presented a novel reperfusion strategy for PCI in patients with STEMI. Our results suggest that VCR had favorable feasibility and clinical safety, with a satisfactory procedural success rate and few MACEs observed during short- to mid-term follow-up. Various aspects of the rational and key advantages of VCR were demonstrated.

First, our technique was shown to be safe and easy to perform using standard laboratory catheterization equipment. Although simultaneous femoral artery access was required, the utilization of vascular closure devices (VCD) could effectively reduce access-site complications and bleeding for patients with acute myocardial infarction undergoing a transfemoral approach [26,27]. VCD was utilized in 26 patients in our study, none of whom encountered vascular complications. Furthermore, the femoral access route allows the possibility of timely insertion of the IABP in STEMI patients who have a high likelihood of developing cardiogenic shock (CS) during or after PCI (delayed CS). Previous randomized studies including the CRISP-AMI trial assessing IABP use in high-risk STEMI patients without CS found that a prophylactic IABP strategy prior to or after PCI showed no significant benefits for angiographic and clinical outcomes but increased the risk of bleeding and stroke [28,29,30]. Additionally, the IABP-SHOCK II trial investigating IABP treatment in patients with AMI complicated with CS showed that IABP prior to or after PCI had no effect on all-cause mortality, recurrent MI, or repeat revascularization at long-term follow-up [31]. These findings indicte that IABP might be redundant for patients without CS and unprofitable for those who have already experienced CS.

Data from previous studies have shown that 4%–12% patients without CS on admission went on to develop delayed CS [32,33,34]. We hypothesized that IABP if timely inserted probably displays better efficacy in patients who have a high likelihood of developing delayed CS (previously, patients would be trapped in their CS status). However, to date, there has been no definite criteria for identification of this subset of patients. Based on the limited data, the coronary flow of the culprit artery during the PCI procedure may be a stronger risk factor associated with development of delayed CS [33,34]. In the present study, three patients encountered a slow-flow phenomenon after the prophylactic balloon was deflated or after stent implantation. Coronary flow recovered well (TIMI grade 3) in the patient who received immediate IABP treatment, whereas the coronary flow failed to reach TIMI grade 3 in the two patients who received delayed IABP insertion. Additionally, in the patient with procedural hypotension, which was detected with the use of the second catheter, the blood pressure recovered well after timely implantation with an IABP. Although the delay of few minutes seems small, we cannot entirely rule out the possibility that the potential beneficial effect of IABP on angiographic and clinical outcomes is confined to patients who received timely support. Dual catheterization with additional femoral access enabled continuous measurement of pressure and simultaneous coronary reperfusion, allowing instantaneous feedback and timely IABP insertion. Our hypothesis should be further evaluated in future multicenter randomized trials.

We conducted reperfusion after suspending the antegrade flow by placing a prophylactic inflated balloon at the site of the index lesion. This method not only enables control the composition, volume, and speed of the infusion liquid, but also avoids repeated inflation and deflation of the balloon in the process of post-conditioning, which might reduce damage to the coronary endothelium and the risk of thrombus dislodgment. Previous trials including PREPARE and PROXIMAL used proximal embolic protection devices to demonstrate that it is safe to interrupt the antegrade flow with balloon pre-dilation and stent implantation during the period of lesion intervention [35,36].

We administered distal reperfusion through an aspiration catheter with sufficient lumen diameter to achieve adequate perfusion, able to provide operators with a convenient method to perform thrombus aspiration if required. Instead of aspiration catheters, distal drug delivery can be performed using over-the-wire balloons (OTWB) or micro-catheters [37,38,39]. It was difficult to perform accurate distal reperfusion by OTWB, and infusion volume and rates were limited due to the narrow caliber of the lumen in micro-catheters. In addition, it was not possible to perform bail-out aspiration using these two devices, in case distal embolization occurred.

Despite various agents such as adenosine, atrial natriuretic peptide, bradykinin, nitrates, nicorandil, and metoprolol having the potential to reduce reperfusion injury, the safety and effectiveness of intracoronary delivery of these agents remains controversial [40]. Therefore, we first selected arterial blood extracted from patients themselves as the perfusate. However, patients’ arterial blood may contain higher volumes of inflammatory factors and metabolic by-products in the STEMI setting, which require counteraction if the blood is to be used as an intracoronary perfusate. Theoretically, heparin can inhibit the abnormal aggregation and adhesion of leukocytes and platelets, prevent the formation of microthrombus, and reduce local inflammation and oxidative stress response [41]. Previous research has demonstrated that it is safe to deliver high volumes (44 mL/min) of heparin solution locally to coronary at very low pressure and long-term effects of local intracoronary delivery of heparin did not show any deterioration of the treated site [20]. Additionally, studies using a microvascular model of thrombosis showed that local delivery of heparin could inhibit thrombus formation without prolonging bleeding [19]. However, as we know, this agent is a “double-edged sword” which can bring benefits and also adverse side effects to patients. Despite no negative evidence [42], we cannot entirely rule out the possibility that a higher concentration of intracoronary heparin might result in intramyocardial hemorrhage (IMH), which may offset the benefits of heparin. Therefore, we used heparin saline (heparin diluted with normal saline, 3000 units of heparin to 500 mL of normal saline) instead of heparin as the component for the agent. The relatively lower concentration of heparin might reduce the risk of IMH as well as the risk of IRI. Additionally, heparin saline might prevent occlusion of the aspiration catheter during reperfusion and might help flush out the metabolic by-products from the microvasculature of the area at risk or the infarct core [43]. Therefore, arterial blood diluted with heparin saline was selected as the final perfusate for VCR.

Direct volumetric measurement of blood flow in selective coronary arteries in conscious humans is a challenging task [44]. We estimated the flow rate and the duration for reperfusion using VCR according to the evidence of previous studies and the experience gained in our center. Otterspoor et al. confirmed the safety and feasibility of selective intracoronary infusion of saline at a rate of 10–30 mL/min in STEMI patients, at room temperature in the ischemia phase (coronary flow was blocked by an inflated OTWB) and at 4 °C in the reperfusion phase (OTWB was deflated). In their study, the method prolonged the occlusion of the coronary artery by approximately 20 min [45]. Similarly, Wang et al. reported that selective coronary artery infusion of saline at 4 °C for 2.5–5 mL/min in the occlusion phase and 10–30 mL/min in the reperfusion phase by use of a modified aspiration catheter was effective, feasible, and safe in conscious STEMI patients. The total infusion time was maintained at 15 min [46]. Hence, the initial reperfusion rate for VCR was set at 20 mL/min and the total infusion duration for 10 rounds. Each round involved approximately 60 s reperfusion time and 15 s pause for diluted artery blood preparation, mimicking the postconditioning procedure. We adjusted the reperfusion flow rate based on heart rate and blood pressure, which were the obvious clinical markers of IRI during the PCI procedure. When heart rate or blood pressure dropped, we paused or reduced the speed of reperfusion until all parameters were again stable. We hypothesized that this quantitative volume-controlled reperfusion strategy could minimize the degree of reperfusion injury. Although there was no control group in the present study, the results observed for the other endpoints suggested that VCR might had the potential ability to decrease IRI. This was especially the case for the extent of MVO, which represents a new frontier in cardioprotection with higher sensitivity for detecting IRI and a closer relation to clinical outcomes than infarct size. We will evaluate the effectiveness of VCR in an upcoming RCT.

It was determined that the protocol generally needs an extra 10–15 min to establish the VCR model and another 10–15 min to perform controlled reperfusion. It remains uncertain whether delayed reperfusion and stenting during the window period could affect outcomes in patients with STEMI. Recently, the DTU-STEMI pilot trial reported that left ventricle unloading using the Impella device with a 30-min delay before reperfusion is feasible in anterior STEMI within a relatively short time [42]. The OXAMI-PICSO study aimed to assess the efficacy of index of microcirculatory resistance (IMR)-guided therapy with pressure-controlled intermittent coronary sinus occlusion (PICSO) before stenting in anterior STEMI, and reported that IMR-guided treatment with PICSO in anterior STEMI is feasible and may be associated with reduced infarct size and improved microvascular function, despite PICSO treatment being associated with a prolonged procedural time (30 min) [47]. Intentionally delaying reperfusion time in STEMI is not recommended, but these two pilot studies challenged to some extent the existing paradigm of STEMI management.

During the follow-up period, one patient underwent TVR because the distal segment of IRA narrowed three months after the index PCI procedure. One possible explanation for this finding was that distal IRA segment in this patient previously had a stenosis of 40–50%. Two patients exhibited TIMI grade 2 at the completion of the PCI procedure. A possible explanation was that these two patients had high risk factors for slow-flow phenomenon, including smoking, higher fibrinogen, and higher total cholesterol and LDL-C levels [48,49]. Furthermore, in terms of the VCR procedure, these cases shared the similar problem that the blocking balloon was not positioned properly at the index lesion. Therefore, selecting the appropriate size of balloon (1:1 ratio with the lesion diameter) was found to be essential for a successful VCR procedure.

The present study has several limitations. First, it should be noted that this was a single-center pilot study with a small sample size and without a control group. Therefore, it may not have adequate statistical power to allow a definitive conclusion. However, our favorable results suggest that it will be worth conducting a further randomized control study to evaluate the safety and effectiveness of VCR in STEMI patients. Secondly, LGE-CMR was not performed in 14 patients for various reasons, including long waiting time for CMR and a limited CMR examination window, inability to cooperate during the CMR examination, claustrophobia in patients, and technical difficulties. Therefore, the ability directly to assess infarct injury was limited. Thirdly, in the present study there was a lack of T2*-weighted cardiac magnetic resonance imaging with higher sensitivity for detecting and quantifying IMH [50]. It remains to be investigated whether patients could obtain net benefits at a lower concentration of heparin.

## 5. Conclusions

This pilot study has demonstrated that the novel reperfusion strategy (VCR) had favorable feasibility and clinical safety in patients with STEMI. Further larger randomized trials are required to evaluate the effectiveness of VCR in STEMI patients.

## Figures and Tables

**Figure 1 jcm-12-00433-f001:**
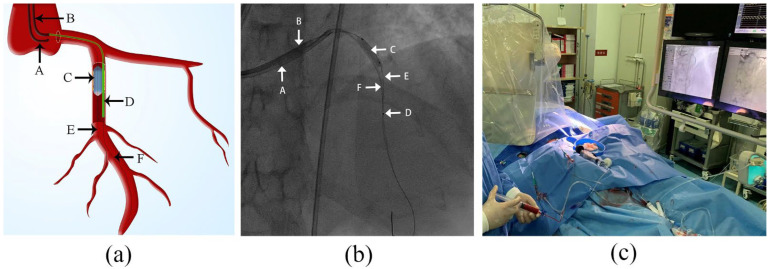
Illustration of volume-controlled reperfusion (VCR) system (**a**) schematic view of VCR system; (**b**) screenshot of VCR system; (**c**) VCR procedure in practice. A: guiding catheter 1; B: guiding catheter 2; C: angioplasty balloon; D: aspiration catheter; E: guide wire 1; F: guide wire 2.

**Figure 2 jcm-12-00433-f002:**
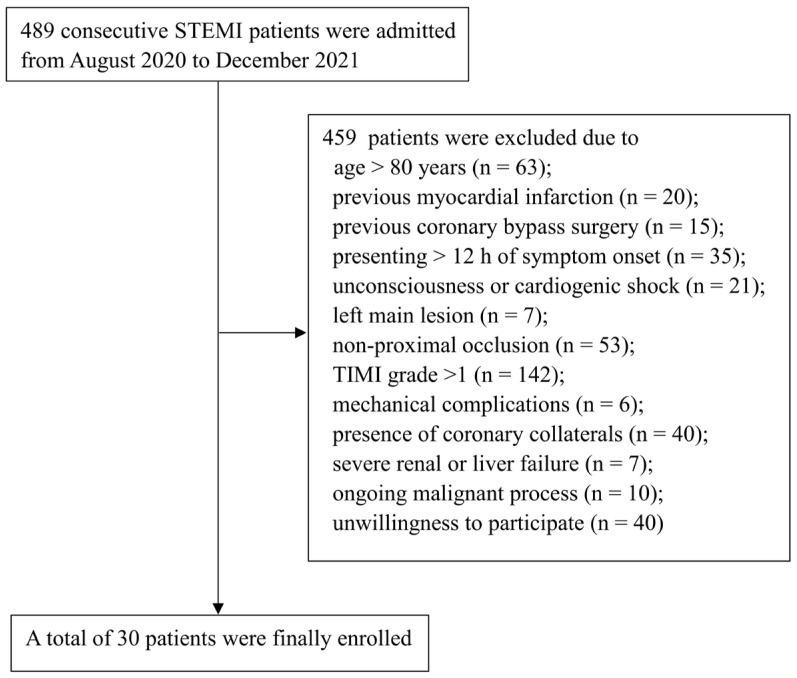
Flow chart of patient selection. STEMI: ST-segment elevation myocardial infarction; TIMI: Thrombolysis in Myocardial Infarction.

**Table 1 jcm-12-00433-t001:** Baseline and clinical characteristics of patients.

Characteristic	Value
Age, years	59.8 ± 12.4
Male gender	26 (86.7)
BMI, kg/m^2^	25.10 ± 3.21
Hypertension	16 (53.3)
Diabetes mellitus	9 (30.0)
Hyperlipidemia	6 (20.0)
Chronic obstructive pulmonary disease	2 (6.7)
Chronic kidney disease	1 (3.3)
Stroke	1 (3.3)
Current smoking	24 (80.0)
Family history of coronary artery disease	3 (10.0)
Admission haemodynamics	
Systolic blood pressure, mmHg	138.1 ± 25.2
Diastolic blood pressure, mmHg	87.7 ± 18.2
Heart rate, bpm	75.4 ± 15.2
Killip classification at admission	
1	18 (60.0)
2	12 (40.0)
Laboratory findings at admission	
TNI, ng/mL	0.10 (0.05, 0.93)
CKMB, ng/mL	3.75 (1.08, 20.35)
BNP, pg/mL	41.90 (17.75, 92.25)
LAC, mmol/L	1.70 (1.40, 2,10)
TC, mmol/L	5.12 ± 0.99
TG, mmol/L	2.62 ± 1.77
HDL-C, mmol/L	0.97 ± 0.18
LDL-C, mmol/L	3.49 ± 0.94
UA, mmol/L	5.75 ± 1.86
CR, umol/L	75.26 ± 23.16
WBC, 10^9^/L	11.28 ± 1.99
HGB, g/L	148.67 ± 18.96
PLT, 10^9^/L	231.00 (202.81, 254.32)
D-dimer, mg/L	0.34 (0.14, 0.72)
FIB, mg/dL	274.23 ± 45.95

BMI: body mass index; TNI: troponin I; CK-MB: creatine kinase isoenzyme MB; BNP: brain-type natriuretic peptide; LAC: lactic acid; TC: total cholesterol; TG: triglyceride; HDL: high-density lipoprotein cholesterol; LDL: low-density lipoprotein cholesterol; UA: uric acid; CR: creatinine; WBC: white blood cells; HGB: hemoglobin; PLT: platelet; FIB: fibrinogen.

**Table 2 jcm-12-00433-t002:** Procedural and angiographic characteristics of patients.

Characteristic	Value
Time intervals, min	
Symptom onset to hospital arrival	194.00 ± 165.40
Hospital arrival to catheterization laboratory	15.70 ± 4.43
Catheterization laboratory to radial access	7.87 ± 2.22
Infarct-related artery	
Left anterior descending coronary artery	17 (56.7)
Right coronary artery	10 (33.3)
Left circumflex coronary artery	3 (10.0)
Multivessel disease	15 (50.0)
TIMI flow grade before PCI	
0	30 (100.0)
1	0 (0.0)
Time for VCR model^※^ establishment, min	12.44 ± 2.03
VCR duration, s	750 (750, 760)
PCI procedure duration, min	44.70 ± 3.23
Mean aortic pressure during VCR, mmHg	100.50 (96.00, 106.00)
Mean heart rate during VCR, bpm	72.00 (61.50, 80.25)
Drug-eluting stent implantation	16 (53.3)
Total stent number (n = 16)	2 (1, 2)
Total stents length, mm (n = 16)	32.50 (22.0, 51.50)
Minimum stent diameter, mm (n = 16)	3.50 (3.00, 3.90)
Drug-eluting balloon implantation	14 (46.7)
Total balloon number (n = 14)	1 (1, 1)
Total balloon length, mm (n = 14)	29.00 (23.80, 32.50)
Minimum balloon diameter, mm (n = 14)	3.00 (3.00, 3.10)
Plain balloon angioplasty	2 (6.7)
Thrombosis aspiration	3 (10.0)
Intra-aortic balloon pump	4 (13.3)
Glycoprotein IIb/IIIa therapy	3 (10.0)
TIMI flow grade < 3 post PCI	2 (6.6)
Non-infarct-related artery with stenosis >70%	15 (50.0)
Intervention for non-infarct-related artery	11 (36.7)
Immediate complete revascularization	5 (45.5)
Staged complete revascularization	6 (54.5)
Doses of heparin, IU	8000.00 (7000.00, 9250.00)
Medication at discharge	
Aspirin	26 (86.7)
Clopidogrel	26 (86.7)
Ticagrelor	3 (10.0)
Statins	26 (86.7)
Beta-blockers	20 (66.7)
Calcium channel blockers	0 (0.0)
ACEI/ARB	15 (50.0)

VCR: volume-controlled reperfusion; TIMI: thrombolysis in myocardial infarction; PCI: percutaneous coronary intervention; ACEI/ARB: angiotensin converting enzyme inhibitors/angiotensinogen type II receptor blockers. ※ VCR model establishment: femoral artery route opened, dual guiding catheter engaged into the target coronary artery, balloon positioned at the site of the index lesion, aspiration catheter advanced to the target segment.

**Table 3 jcm-12-00433-t003:** Feasibility endpoints.

Variable	Value
Procedural success	28 (93.3)
Successful completion of operation process	30 (100.0)
Maintaining stable hemodynamics during the entire infusion	30 (100.0)
Achieving a TIMI grade 3 post-procedure	28 (93.3)

TIMI: thrombolysis in myocardial infarction.

**Table 4 jcm-12-00433-t004:** Safety endpoints.

Variable	Value
All-cause death	0 (0.0)
Major vascular complications	0 (0.0)
Major adverse cardiac events	1 (3.3)
Cardiac death	0 (0.0)
Myocardial reinfarction	0 (0.0)
Target vessel revascularization	1 (3.3)
Heart failure	0 (0.0)

**Table 5 jcm-12-00433-t005:** Other endpoints.

Variable	Value
Complete ST-segment resolution at 60 min after PCI	21 (70.0)
Cardiac biomarkers peak post-PCI	
CKMB peak (ng/mL)	176.85 (97.05, 289.7)
TNI peak (ng/mL)	184.78 (56.04, 258.45)
BNP peak (pg/mL)	327.00 (126.75, 567.00)
LVEF ≥ 50% within 3 days after PCI	27 (90.0)
LGE-CMR within 7 days after PCI (n = 16)	
Infarct size, g	33.77 (24.45, 44.20)
MVO mass, g	0.64 (0.14, 1.82)

PCI: percutaneous coronary intervention; CK-MB: creatine kinase isoenzyme MB; TNI: troponin I; BNP: brain-type natriuretic peptide; LVEF: left ventricular ejection fraction; LGE-CMR: late gadolinium-enhanced cardiovascular magnetic resonance; MVO: microvascular obstruction.

## Data Availability

The data that support the findings of this study are available from the corresponding author, Le-Feng Wang, upon reasonable request.

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
