# Peer review of "A Novel Reperfusion Strategy for Primary Percutaneous Coronary Intervention in Patients with Acute ST-Segment Elevation Myocardial Infarction: A Prospective Case Series"

_jcm, 2023, doi:10.3390/jcm12020433_

Round 1

Reviewer 1 Report

The authors have tried to show that the proposed method is superior. However, this article lacks scientific evidence. At least, in order to claim efficacy, at least a retrospective study should have been conducted. It is difficult to distinguish whether results without comparison with a control group are due to chance. I'd like to recommend the authors to conduct an analysis using EMR to create the scientific proof.

Reviewer 2 Report

Ji-Fang He and co-authors reported the feasibility study of a novel technique to prevent reperfusion injury in patients with myocardial infarction (MI) undergoing PCI. Specifically, they assessed the acute and midterm efficacy and safety of the VCR procedure in a group of thirty patients with MI.

Comments.

1.       The introduction does not properly introduce the novel method of post-conditioning. No typical rationale for new development is provided, e.g. what is lacking in the existing post-conditioning techniques, how this novel technique may address these needs, what are the general/specific advantages of this novel approach over the existing ones.

2.       The authors defined the technique as “volume-controlled reperfusion (VCR)”; however, no proper details are provided how the volume of reperfusion is controlled and what are the benefits of such control.

3.       The Title defines the type of research as a case series study, i.e. retrospective and observational thereby providing the lowest quality of evidence. However, Methods section describes the enrollment, inclusion/exclusion criteria, primary/secondary endpoints typical for a clinical trial, i.e. prospective interventional study generating evidence of much higher quality.

4.       The selection of primary endpoint is only necessary if one is concerned about the statistical power of the results/sufficiently sized study sample to confirm/reject the hypothesis of the study, which is typically in confirmatory studies (several endpoints introduces the multiplicity problem - necessity to lower the level of statistical significance for each endpoint, another problem is that different endpoints generally require different sample size to reach statistical power, therefore, one of them should be selected as primary to simplify the situation and provide a single definitive conclusion regarding the hypothesis of the study). Obviously, this is not the case with the pilot/feasibility study

5.       If the primary endpoint is still defined, the sample size should be properly calculated and followed by proper enrollment to provide a robust assessment of the endpoint. With in-hospital mortality for MI typically within several percent what sense does it make to assess it as a primary endpoint in the sample of 30 cases (a single death may produce a two-fold change in the endpoint value)

6.       The key factor advancing reperfusion injury from moderate to severe is hemorrhage then Heparin seems suboptimal agent to reduce the reperfusion damage.

7.       It should be discussed what potential benefits of this approach may outweigh the obvious minuses of additional femoral access. Currently, it is quite unclear why someone should consider this approach instead of existing techniques of post-conditioning.

Round 2

Reviewer 1 Report

This paper has been much improved. I am happy to see that you clarified that this is a pilot study.